# Evaluation of the Predictive Ability and User Acceptance of Panoramix 2.0, an AI-Based E-Health Tool for the Detection of Cognitive Impairment

**Sonia Valladares-Rodríguez** [1], **Manuel J. Fernández-Iglesias** [2] , **Luis E. Anido-Rifón** [2,]*  
**and Moisés Pacheco-Lorenzo** [2]

1    Artificial Intelligence Department, Universidad Nacional de Educación a Distancia, 28012 Madrid, Spain  
2    atlanTTic, University of Vigo, 36310 Vigo, Spain  
\*    Correspondence: lanido@det.uvigo.es

**Abstract:** The high prevalence of Alzheimer-type dementia and the limitations of traditional neuropsychological tests motivate the introduction of new cognitive assessment methods. We discuss the validation of an all-digital, ecological and non-intrusive e-health application for the early detection of cognitive impairment, based on artificial intelligence for patient classification, and more specifically on machine learning algorithms. To evaluate the discrimination power of this application, a cross-sectional pilot study was carried out involving 30 subjects: 10 health control subjects (mean age: 75.62 years); 14 individuals with mild cognitive impairment (mean age: 81.24 years) and 6 early-stage Alzheimer's patients (mean age: 80.44 years). The study was carried out in two separate sessions in November 2021 and January 2022. All participants completed the study, and no concerns were raised about the acceptability of the test. Analysis including socio-demographics and game data supports the prediction of participants' cognitive status using machine learning algorithms. According to the performance metrics computed, best classification results are obtained a Multilayer Perceptron classifier, Support Vector Machines and Random Forest, respectively, with weighted recall values >= 0.9784 ± 0.0265 and F1-score = 0.9764 ± 0.0291. Furthermore, thanks to hyper-parameter optimization, false negative rates were dramatically reduced. Shapley's additive planning (SHAP) applied according to the eXplicable AI (XAI) method, made it possible to visually and quantitatively evaluate the importance of the different features in the final classification. This is a relevant step ahead towards the use of machine learning and gamification to early detect cognitive impairment. In addition, this tool was designed to support self-administration, which could be a relevant aspect in confinement situations with limited access to health professionals. However, further research is required to identify patterns that may help to predict or estimate future cognitive damage and normative data.

**Keywords:** early detection; cognitive impairment; gamification; machine learning algorithms; eXplicable AI (XAI); non-intrusive assessment

## 1. Introduction

Dementia currently affects more than 50 million people worldwide. It is estimated that by 2030 more than 75 million people will suffer from this disease and that this figure will triple by 2050 [1]. These data show the high prevalence of this condition, as well as its social impact, since it affects such a large population. Specifically, Alzheimer's disease is the most common cause of dementia and may contribute to 60–70% of all cases. Throughout, the World Health Organization recognizes dementia as a public health priority and makes a global call against it [2]. In addition, the likelihood of developing dementia increases with advancing age, so that the aging of Western populations is also another factor that will further increase its prevalence.

The diagnostic methods to detect such a medical condition, from a cognitive point of view, are the so-called neuropsychological tests [3], such as the Mini-mental State Examination (MMSE) for dementia or Reisberg's Global Deterioration Scale (GDS). GDS is a dementia diagnosis and rating system used to classify patients according to the experienced stage of mild cognitive impairment or dementia, regardless of cause [4]. These tools perform a neuropsychological assessment addressing certain areas, such as memory, language, attention or visuospatial ability. However, despite their normative value, these tools have some limitations, such as being affected by the white coat effect, as they are generally perceived by users as intrusive and alien tools [5]; providing a late diagnosis [6]; notoriously lacking ecological validity [7,8]; and being dependent on confounding variables (e.g., age, educational level [9], practice effect or testing effect [10,11], etc.).

For all these reasons, alternative cognitive assessment mechanisms were explored in recent years, including the digitization of classic tests [12], gamification [13] or virtual reality [14–21], among others. In particular, our proposal is based on the introduction of serious games, immersive virtual reality and artificial intelligence (AI), more specifically machine learning algorithms [22,23]. Serious games are "games that do not have enjoyment, entertainment or fun as their primary purpose" [24]. It is possible to find in the literature several proposals to perform neuropsychological assessment using videogames addressing attention [20,25,26], working memory [27,28] and executive functions [29,30], among others. Gamification techniques offer significant advantages [5] to assess the cognitive status of an individual, as they are more ecological, reproducing real-life situations, and are not perceived as an intrusive tool. However, more research is still needed to build reliable and valid serious game-based tests ready to be used in clinical neuropsychological assessment [22]. For this, an adequate psychometric validation of existing solutions is required [31,32].

This paper discusses the effectiveness of Panoramix 2.0, a battery of three serious games, for the detection of cognitive impairment in older adults supported by machine learning artificial intelligence (AI) algorithms. Originally, this e-health tool, named Panoramix, was designed by the authors as a digital test consisting of 6 games, which once psychometrically validated served as the foundation of Panoramix 2.0, a new 3-game version with improved usability, support for additional cognitive tasks and better scalability. This new version focuses on assessing the most predictive cognitive areas identified in previous studies for Mild Cognitive Impairment (MCI) [33,34], namely episodic memory [35–37], semantic memory and procedural memory. In addition, this tool was designed to support self-administration. Users may play with the three games in the battery on their own, and all data collected would be submitted online to a health facility to be further analyzed by health professionals. This is a relevant feature in confinement situations such as the ones consequence of the coronavirus disease of 2019 (COVID-19) pandemic.

Throughout this paper, we will present the results of a pilot experiment, described in Section 2, whose main objectives are (1) to validate the predictive ability of the new version of the digital test to discriminate healthy people from those suffering from cognitive impairment and (2) to measure the degree of acceptance by the participants of this new tool. The results of this pilot are presented in Section 3 and discussed in Section 4. Finally, concluding remarks are offered in Section 5.

## 2. Materials and Methods

### 2.1. Selection and Description of Participants

The distribution of participants according to main statistics is collected in Table 1. Sociodemographic characteristics include gender, age, educational level (0 = illiterate; 1 = ability to read and write; 2 = primary; 3 = secondary; 4 = high school; 5 = vocational training; 6 = university), exercise level and socialization level, all of them based on a 5-point Likert scale ranging from 1 (never) to 5 (always); and finally, chronic treatment (0 = no; 1 = yes).

**Table 1.** General description of the population sample utilized.

| Variables | | Sample n (%) Mean (SD) |
|---|---|---|
| Gender | Female | 22 ($\pm$0.73) |
| | Male | 8 ($\pm$0.27) |
| | | Mean (SD) |
| Age (65+ years) | n = 30 | 78.64 ($\pm$7.23) |
| Educational level | | 2.42 ($\pm$1.2) |
| Exercise level | | 3.57 ($\pm$0.99) |
| Socialize level | | 4.17 ($\pm$1.05) |
| Chronic treatment | | 0.70 ($\pm$0.36) |

(SD = standard deviation).

A total of 30 subjects (22 women and 8 men) from southern Galicia (Spain) participated. The sample was divided into three groups: (1) 10 persons without cognitive impairment or healthy control (HC) group (average age of 75.62 years); (2) 14 patients experiencing mild cognitive impairment (MCI, average age of 81.24 years); and finally, 6 patients suffering from incipient Alzheimer's disease (AD, average age of 80.44 years).

Participants were provided by the Galician Association of Relatives of Alzheimer and other Dementia Patients (AFAGA), and social and health professionals of this organization provided participants' diagnosis or cognitive status. All of them meet the inclusion criteria, namely being over 65 years of age, from a semi-urban Galician environment and having a proactive attitude towards the pilot. In addition, none of them suffered from an advanced stage of dementia, severe disability or technological phobia; aspects that would exclude them from the experiment. All participants read and gave written informed consent before participation in this study. The study procedure was approved by the Autonomic Research Ethics Committee of Galicia (Spain).

*2.2. Design of the Study*

2.2.1. Procedure

The baseline for the pilot experiment was established by means of an initial interview, in which they were administered several classic tests, were asked to complete an initial perception survey, and played the three games in the new version of the digital test. Then, a period of 1 month was left between the tests to avoid the test–retest effect, and the administration of the 3 serious games was repeated again. The pilot was completed with a final questionnaire on the experience and participants' perception about these games. All sessions were carried out in regular cognitive sessions in AFAGA premises. This guaranteed a relaxed, non-intrusive environment, close to the subjects' routines, that is, an environment as ecological as possible to carry out a cognitive evaluation experiment.

2.2.2. Development and General Information Questionnaires

Our team developed and administered two questionnaires to collect information on personal and family life, education, basic medical and neuro-psychology aspects and videogames preferences. The questionnaires were used to verify that all participants met the inclusion and exclusion requirements; to collect cross-sectional information about the main confounding variables [38] (e.g., age, gender, educational level, regular physical exercise and level of socialization, among others); and finally to collect relevant data about participants' previous technological attitude according to the Technology Acceptance Model (TAM), that is, perceived usefulness, perceived ease of use and perceived enjoyment.

### 2.2.3. Traditional Neuropsychological Test

The GDS neuropsychological screening test was selected with a double purpose: to obtain a ground truth or golden standard to correlate with data extracted from the digital test and to validate or diagnose the cognitive status of participants in the study [39,40]. Participants affected by dementia also had a clinical diagnosis. The GDS test was administered to all participants.

### 2.2.4. Digital Neuropsychological Test: Panoramix

As pointed out above, the authors developed a digital test based on gamification, virtual immersion and artificial intelligence, to perform a neuro-psychological assessment using exclusively gaming data. The design approach followed was based on replicating real-life situations using virtual reality for the purpose of achieving an ecological tool.

The resulting artifact, Panoramix, consists of a main game that assesses episodic memory—as this is a clear predictor of MCI and AD—called Episodix and two additional games. Episodix implements a more ecological version of the California Verbal Learning Test (CVLT) where, instead of learning and recalling shopping lists, a virtual walk through a medium-sized town is emulated. Along the walk, objects are presented—in visual and audio format—integrated naturally in the environment. Episodix consists of the same phases as CVLT, that is, stimuli presentation and remembering of objects. Players have to learn and recall as many objects as possible across several phases. Variables captured from Episodix include the ones in the classical test (e.g., yes/no recognition, free recall, short and long delayed recall, recency, primacy, semantic clustering, response to inhibitions, etc.), and also variables obtained transparently from user interactions (i.e., hits, repetitions, guesses, omissions, errors, and total time spent in each phase).

Through the two additional games it is possible to evaluate semantic memory by means of Semantix, which emulates the Pyramids and Palm Trees test [41], and procedural memory and executive functions by means of Procedurix, which reproduces the Rotor Pursuit test [42]. Semantix consists of 52 sets of 3 images corresponding to the stimulus given, the target stimulus, and the distraction stimulus. The subject tries to match the stimulus to one of the other two images. Semantic memory is assessed from the analysis of the total playing time, number of hits, number of errors, number of omissions, and the total number of right answered chips. Finally, Procedurix consists of tracking a rotating circle as accurately as possible using the index finger on a touch-sensitive display. The variables used to assess procedural memory include playing time, deviation from the element to be tracked, initial response time, and total time on track.

The games in the digital test were developed in Unity to be played by means of a regular touch device (e.g., smartphone, tablet). Data management was facilitated by a Fast Healthcare Interoperability Resources (FHIR) [43] server provided by the Gatekeeper project through an open call.

### 2.3. Analytical Algorithms

Different machine learning (ML) algorithms were utilized, all of them widely used in medical research [44–46]: (i) a Logistic Regression linear model (LR), (ii) a Support Vector Machines (SVMs) model, which is based on hyperplanes, (iii) a Random Forest (RF) ensemble method, (iv) the AdaBoost classifier (ADB), (v) k-Nearest Neighbors (kNN), and finally (vi) Multi-layer Perceptron classifier (ANN), based on neural networks. The rest of the methods (i.e., RF, ADB and KNN) are tree-based algorithms. Moreover, we used k-fold cross validation to test and evaluate the aforementioned algorithms splitting data for this procedure according to 80–20 composition (i.e., 80% for training and 20% for testing the resulting model).

Quantitative metrics were also utilized as performance indications:

- F1-score as a measure of a test's accuracy.
- The ratio of correct predictions vs. the total number of input samples.

- Precision, also called positive predictive value, to compute the fraction of relevant instances among the retrieved instances.
- Recall, also known as sensitivity, to compute the fraction of relevant instances retrieved.
- Specificity or true negative rate, that refers to the probability of a negative test, conditioned on truly being negative.

Metrics above were computed according to three models: (1) Micro, that computes a global average of metrics: (2) Macro, which treats all classes equally regardless of their support values; and (3) Weighted, where metrics are calculated by taking the mean per-class. Essentially, this refers to the proportion of each class's support relative to the sum of all support values. In our studio, as we are working with an imbalanced dataset where all classes are equally important, using the macro average would be a good choice as it treats all classes equally.

The need to improve the transparency and explainability of AI-based decisions is broadly discussed in the literature [47,48] and many different methods were proposed over the years [49]. According to our specific domain and the tasks to be performed (i.e., to detect MCI) explainability may become a fundamental requirement to offer a suitable AI solution. Therefore, Shapley's additive planning (SHAP) applied according to the eXplicable AI (XAI) methodology was selected to explain and interpret the classification decisions of the ML models, together with the relevance of individual features to detect the cognitive state.

Finally, hyper-parameter tuning is performed to improve the results obtained [50,51]. To this end, the model's hyper parameters are fine-tuned to maximize the resulting recall without significantly affecting precision. The main hyper-parameters tuned are C—the regularization strength—, gamma—to control the distance of influence of a single training point, with low values of gamma indicating a large similarity radius, which in turn results in more points being grouped together—, kernelization—the application of functions so that features become linearly separable—, and class-weighting—related with the degree of balancing among classes. Data analytics was performed with the Scikit-Learn [52] machine learning library under a Python ecosystem.

## 3. Results

The ability to predict mild cognitive impairment of Panoramix 2.0, the updated and improved version of the Panoramix cognitive battery, is discussed below. This predictive ability and its acceptability by the target population were analyzed for the sample (n = 30) and for the defined cognitive groups, namely HC (n = 10), MCI (n = 14) and AD (n = 6).

### 3.1. General and Cognitive Participants' Characteristics

Participants were characterized according to the parameters below, grouped by cognitive group (cf. Table 2):

- Educational level (i.e., years of education): 3.00 for people without cognitive impairment; 2.14 for the MCI group; and 2.83 mean years of education for people with AD.
- Exercise level (i.e., 5-point Likert scale: 1 (nothing) to 5 (a lot)): 3.50 for controls; 3.43 for the MCI group; and 4.50 for the AD group.
- Socialization level (i.e., 5-point Likert scale: 1 (no social interaction) to 5 (frequent social interaction): 4.20 for HC subjects; 4.00 for participants with MCI; and 4.67 for people affected by AD.
- GDS (cut-off score: 1: HC; 2–3: MCI and >3 AD). In this case, the average score for controls was 1.00; 2.50 for subjects with MCI, and 3.67 for AD participants.

**Table 2.** Participants' general and cognitive characteristics by cognitive state.

| GENERAL SUBJECTS' CHARACTERISTICS | HC (n = 10) | MCI (n = 14) | AD (n = 6) |
|---|---|---|---|
| | Mean (SD) | Mean (SD) | Mean (SD) |
| Age | 75.62 (±6.69) | 81.24 (±5.72) | 80.44 (±3.39) |
| Educational Level | 3.00 (±1.13) | 2.14 (±0.66) | 2.83 (±0.98) |
| Exercise level | 3.50 (±0.71) | 3.43 (±0.65) | 4.50 (±0.55) |
| Socialization level | 4.20 (±0.64) | 4.00 (±0.00) | 4.67 (±0.52) |
| GDS | 1.00 (±0.00) | 2.50 (±0.52) | 3.67 (±0.52) |

Notes: SD = standard deviation. General characteristics: age; chronic treatment (i.e., 0 = no; 1 = yes); educational level (i.e., 0 = illiterate; 1 = ability to read and write; 2 = primary school; 3 = secondary school; 4 = high school; 5 = vocational training; 6 = university), exercise level and socialization level, all of them based on a 5-point Likert scale: 1(never) to 5 (always). Cognitive characteristics: GDS cut-off score: 1: HC; 2–3: MCI and >3 AD likely.

### 3.2. Prediction of Cognitive Impairment Using Machine Learning Classifiers

Table 3 below collects the main metrics obtained to evaluate the classification capabilities of the applied algorithms. The main macro metrics computed were:

- F1-score: 1.0000 for SVM, ANN and KNN. The worst result obtained was 0.8510 in the case of ADB.
- Accuracy: a value greater than 0.9150 was obtained for all algorithms. The maximum value of 1.0000 was obtained for SVM, ANN and KNN.
- Recall or sensitivity: 1.0000 for SVM, ANN and KNN. Again, the worst result was obtained in the case of ADB (0.8000).

**Table 3.** Main classification metrics without hyper-parameters.

| ML Algorithm | Accuracy | Sensitivity (Recall) ↓ | Specificity | Precision | F1-Score |
|---|---|---|---|---|---|
| SVM | 1.00 (±0.000) | 1.00 (±0.000) | 1.00 (±0.00) | 1.00 (±0.00) | 1.00 (±0.00) |
| ANN | 1.00 (±0.000) | 1.00 (±0.000) | 1.00 (±0.00) | 1.00 (±0.00) | 1.00 (±0.00) |
| KNN | 1.00 (±0.000) | 1.00 (±0.000) | 1.00 (±0.00) | 1.00 (±0.00) | 1.00 (±0.00) |
| LR | 0.98 (±0.027) | 0.96 (±0.027) | 1.00 (±0.00) | 1.00 (±0.00) | 0.98 (±0.027) |
| RF | 0.95 (±0.026) | 0.84 (±0.023) | 1.00 (±0.00) | 1.00 (±0.00) | 0.91 (±0.025) |
| ADB | 0.91 (±0.025) | 0.80 (±0.022) | 0.96 (±0.027) | 0.90 (±0.025) | 0.85 (±0.024) |

To stabilize and improve the prediction results, the ML algorithms were trained again, but in this case applying the best configuration of hyper parameters found (cf. Table 4). For the first case, a full recall or sensitivity rate was achieved, while for SVM and RF, values obtained were 0.98 and 0.97, respectively. In general, the classification results improved for all algorithms, compared with the initial run without hyper parameters. Thus, for the tree models, the best configuration was as follows:

- Multi-layer perceptron classifier (activator: 'relu', hidden_layer_sizes = (50, 50), solver = 'lbfgs').
- Support Vector Machine (C = 10, random_state = 42).
- Random Forest Classifier (bootstrap = False, n_jobs = −1, random_state = 42).

**Table 4.** Main classification metrics with hyper-parameters.

| ML Algorithm | Recall Weighted ↓ | | F1-Score Weighted | |
|---|---|---|---|---|
| | Mean | SD | Mean | SD |
| ANN | 1.0000 | ±0.0000 | 1.0000 | ±0.0000 |
| SVM | 0.9889 | ±0.0222 | 0.9873 | ±0.0254 |
| RF | 0.9784 | ±0.0265 | 0.9764 | ±0.0291 |
| LR | 0.9784 | ±0.0265 | 0.9764 | ±0.0291 |
| KNN | 0.9673 | ±0.0268 | 0.9873 | ±0.0254 |
| ADB | 0.8234 | ±0.0660 | 0.8808 | ±0.0397 |

Finally, in Figures 1 and 2 below, it can be observed the results of applying the SHAP visual applicability technique on the importance of the features in cognitive state classification models. In the case of subjects with MCI and HC, the most determinant variable is the duration of the long-term recognition phase. In the case of people affected by AD, the most relevant variable is age, followed by educational level and the success rate in the long-term recognition phase.

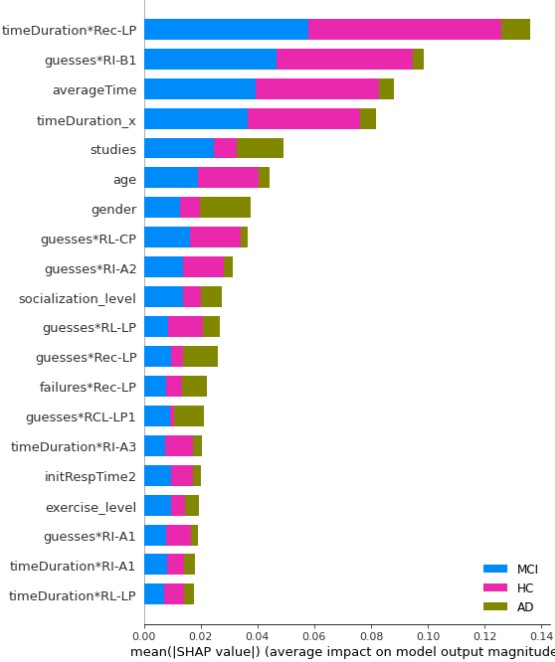

**Figure 1.** SHAP values about importance's features for cognitive classes.

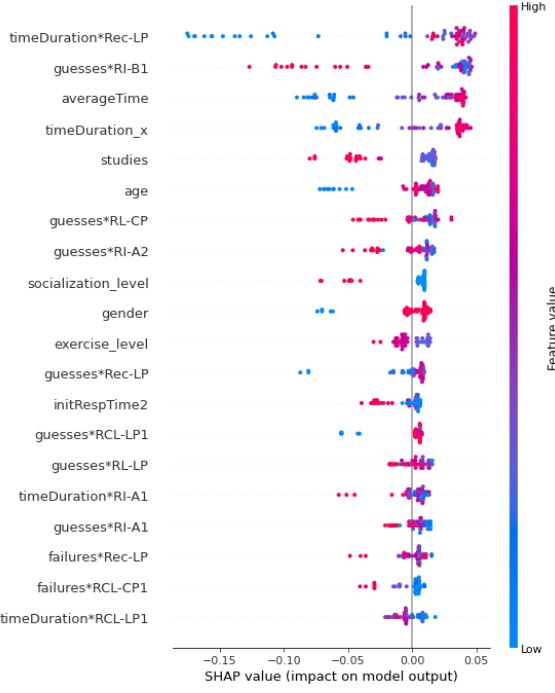

**Figure 2.** SHAP values indicating the relevance of features for participants with MCI (variables are ranked from higher to lower relevance; the horizontal axis shows whether the effect of the corresponding value is associated with a higher or lower prediction; colors shows whether variables' original values were high (in pink) or low (in blue) for that observation).

### 3.3. Participant's Differences in User Experience of Cognitive Games

Participants expressed their perception about the digital test discussed in this work. After using each of the three games, they answered a questionnaire following the TAM model (cf. Table 5). The main results indicate that they liked the tool: 4.55 on average for the HC group; 4.13 for the MCI and 3.99 for people with AD, in a 5-point Likert scale. As for the ease of use, the results vary from an average of 4.62 for the HC group to 4.19 for people affected with AD. Finally, all the cognitive groups indicated that they would use this test again, with a rating of at least 4.23 out of 5.

**Table 5.** Participants' perception about Panoramix.

| SUBJECTS' PERCEPTION | HC (n = 10) | MCI (n = 14) | AD (n = 6) |
|---|---|---|---|
|  | Mean (SD) | Mean (SD) | Mean (SD) |
| P1. I liked this game very much | 4.50 (±0.02) | 4.13 (±0.78) | 3.99 (±0.48) |
| P2. I find this game useful to exercise my memory | 4.63 (±0.02) | 4.36 (±0.83) | 4.06 (±0.49) |
| P3. I found the instructions clear | 4.66 (±0.02) | 4.69 (±0.88) | 4.20 (±0.51) |
| P4. I find this game easy to play | 4.54 (±0.02) | 4.53 (±0.85) | 3.84 (±0.46) |
| P5. I find this game easy to control using my fingers | 4.62 (±0.02) | 4.63 (±0.87) | 4.19 (±0.51) |
| P6. This game is good for my memory, and I would keep using it | 4.61 (±0.02) | 4.28 (±0.81) | 4.23 (±0.51) |

Note: SD: standard deviation. TAM Questionnaire perception about Panoramix; all of them based on a 5-point Likert scale: 1(strongly disagree) to 5 (strongly agree).

## 4. Discussion

This work studies the ability to identify mild cognitive impairment through the new version of Panoramix, a three game-based battery to evaluate cognitive impairment in senior adults, using supervised machine learning techniques. Apart from the reduction from six to three games keeping the predictive capabilities of the original batteries, games were updated with improved usability features, cognitive tasks and scalability. This new version focuses on evaluating the most predictive cognitive areas [33,34] to identify MCI, such as episodic memory [35–37], semantic memory and procedural memory. To do this, a new pilot experiment was carried out to assess the predictive capacity of MCI and the evaluation of the new tool by participating users. A total of 30 individuals completed the pilot study with a two-fold objective: (1) to validate the predictive capacity of the new version of the digital test to discriminate healthy people from those suffering from cognitive impairment and (2) to measure the degree of acceptance by the participants of this new tool.

The sample selected for this cross-sectional study met the established inclusion criteria (i.e., participants being over 65 years of age, from a semi-urban Galician environment and having a proactive attitude towards the experience) and no participants opted out or were excluded due to technological phobia. The pilot's distribution in terms of age and educational level (i.e., all participants had primary or secondary education) is within the parameters of the Galician society, with a slight imbalance in terms of gender by cognitive class, although balanced overall. The level of sociability and physical exercise stands out, especially in the group of people affected by AD, 4.50/5 and 4.67/5, respectively. These aspects were regularly addressed at the cognitive workshops carried out by the AFAGA association, in accordance with the non-pharmacological recommendations for improving cognitive reserve. Finally, the assessment used the GDS scale as a golden standard, as participants were initially classified according to the cut-off scores of this test: the mean score for controls was 1.00, 2.50 for subjects with MCI, and 3.67 for AD participants.

Firstly, in relation to the predictive ability, classification results provided by three mainstream machine learning algorithms—LR, RF, SVM, ANN, ADB and KNN—were analyzed using a 10-fold stratified cross validation procedure [53]. After refinement of the initial classification results (cf. Table 3) by applying hyper-parameters and maximizing

the recall weighted rate to stabilize the results (cf. Table 4), best results were obtained with ANN, SVM and RF, respectively, with weighted recall values >= 0.9784 ± 0.0265 and F1-score = 0.9764 ± 0.0291, in line with previous supervised classification studies targeting the detection of cancer [54] and dementia [55]. It should be noted that the remaining algorithms, LR and RF, performed very similarly. In short, the changes made to the new version of Panoramix do not compromise its prediction capabilities when compared to previous validations [31,32]. Thus, these results indicate that Panoramix is a high-quality tool to predict cognitive impairment, both according to the fraction of relevant instances that were retrieved (i.e., recall), as well as from the point of view of (the harmonic average of) precision and sensitivity (i.e., F1).

It should be noted that the XAI techniques used made it possible to visually and quantitatively evaluate the importance of the different variables/features for the final classification. In the case of subjects with MCI and HC, the most informative variable is the duration time of the long-term recognition phase in the Episodix game, which is one of the most demanding cognitive recall tasks in Panoramix 2.0. In contrast, in the case of AD-affected individuals, the most informative variable is age, in line with major medical paradigms of higher prevalence of cognitive impairment the older the age [2], in addition to the level of education and correct guesses in the long-term recognition phase.

Secondly, in relation to the evaluation of the digital test, the participants expressed their perception through a questionnaire following the TAM model (cf. Table 5). In general, this perception was highly positive for all cognitive groups, although it was somewhat lower for people affected by AD. In general, the acceptance of this test was very good, as they liked the tool (4.55/5 on average for the HC group; 4.13/5 for MCI's and 3.99/5 for people with AD). Another relevant element is the test's perceived ease of use, as the ratings varied from 4.62/5 for the HC group to 4.19/5 for people affected with AD. Finally, all the different cognitive groups indicated that they would use this digital tool again with a rating of at least 4.23 out of a total of 5. This shows the absence of both rejection attitude, to the test and the white coat effect, which is described in the case of classic tests.

In relation to the limitations of this study, it is worth mentioning that to achieve more representative results, a larger sample, including subjects adequately distributed according to age and gender groups, and involving at least two countries to consider cultural or linguistic factors, would be necessary. In addition, a larger sample would allow the acquisition of a larger dataset, which would mitigate the chances of overfitting data, which were not negligible for the current dataset despite the use of cross validation techniques to mitigate it. All in all, this paper corroborate the initial results discussed in [23,31], and also shows that Panoramix is a valid tool to discriminate between MCI, AD and cognitively-unimpaired individuals by means of an ensemble of ML classifiers [44,46,56] and performance metrics obtained.

## 5. Concluding Remarks

Panoramix 2.0, the new version of the Panoramix digital test for the early detection of MCI consisting of three games instead of six, is confirmed to serve to discriminate healthy subjects from those with cognitive problems (e.g., MCI or AD) in a non-intrusive, environmentally friendly and frustration-free environment, which is especially important for participants with cognitive limitations. By collecting a reduced set of informative features along a 40-minute playing session, it is possible to correctly discriminate healthy individuals from subjects experiencing mild cognitive impairment. The analysis is carried out along a non-invasive and friendly activity, positively perceived by senior adults, in a way totally transparent to them. This is a promising step in the use of gamified digital tests in this area. In any case, further research is needed, involving a larger and more diverse participant sample, to obtain normative data to validate this new neuropsychological assessment system according to clinical standards.

**Author Contributions:** All authors have contributed equally. All authors have read and agreed to the published version of the manuscript.

**Funding:** This research was partially funded by (1) the Gatekeeper project under its 1st Open Call, which received funding from the European Union's Horizon 2020 research and innovation program under grant agreement number 857223, and (2) grant PID2020-115137RB-I00 funded by the Spanish Ministry of Science and Innovation (DOI 10.13039/501100011033).

**Data Availability Statement:** The data presented in this study are available on request from the corresponding author. The data are not publicly available due to privacy restrictions.

**Acknowledgments:** We will like to thank the support of the Association of Relatives of Alzheimer's Patients and other Dementias of Galicia (AFAGA), to facilitate the selection of participants among its members for the present pilot study, and DataSalus SL for its support with pilot implementation.

**Conflicts of Interest:** The authors declare no conflict of interest.

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
