# Peer review of "Evaluation of the Predictive Ability and User Acceptance of Panoramix 2.0, an AI-Based E-Health Tool for the Detection of Cognitive Impairment"

_electronics, doi:10.3390/electronics11213424_

Round 1
Reviewer 1 Report
This paper entitled “Evaluation of the predictive ability and user acceptance of Panoramix 2.0, an AI-based e-health tool for the detection of cognitive impairment” by Valladares-Rodríguez et al. evaluated the cognitive impairment predictive ability and user acceptance of Panoraminx 2.0.
This study validated a video game and its predictive ability of cognitive impairment. Authors are the very first one to evaluate this game. They have reported similar games with similar methods. This makes the novelty of this study very limited.
The major short of this paper is the writing quality of the paper is not ready to publish. There are some of the concerns will be addressed as follows:
1 Authors didn’t mention why they particular interested in Panoramix 2.0?
2 SD are missing in table 2 and 3.
3 Error bars and figure legends in figure 1 are missing.
4 Abbreviations are used without definition.
Author Response
Rebuttal (electronics-1933078):
Title:
Evaluation of the predictive ability and user acceptance of Panoramix 2.0, an AI-based e-health tool for the detection of cognitive impairment
1 Reviewer 1:
The work presented in this manuscript is primary on using machine learning method for early detection of Alzheimer‐type dementia. The work can be included for possible publication in this journal due to the following reasons:
(1) Although the machine learning method used in the subject focus of interest is not new, it does fulfill the technology gap of traditional neuro-psychological tests, which are time consuming and inaccurate.
(2) The method and test procedure are clearly described and the concluding remarks are supported by the results, of which results indicate the work does meet archival journal merits of enhancement of exiting knowledge in the cognitive assessment method for the Alzheimer‐type dementia.
Thank you for your comments. The reasons that you pointed out are in line with the authors’ motivations to submit this manuscript to be considered for publication in Electronics.
Where are areas for improvement:
(1) Although the present results are promising, it is suggested to investigate more on the sensitivity of information questionnaires and ML algorithms used in the digital assessment. Will the results be different with different algorithm such as decision tree, AI, ANN, etc?
The manuscript collects the outcomes of a research work that investigated several machine learning algorithms and fine-tuned information questionnaires to select the most appropriate solution for our purposes, namely maximizing recall (i.e., the amount of successfully classified subject in the sample), and consequently the number of false negatives. This was our priority because classifying an individual experiencing cognitive decline as a healthy individual means delaying intervention for that individual. Early intervention is key when dealing with cognitive conditions.
We have included in the revised version of the manuscript the result obtained with different algorithms, more specifically ANN, SVM, RF, LR, KNN and ADB (cf. Tables 3 & 4 in the revised version. In this way, we offer a broader perspective of the underlying research works about the performance of different ML techniques.
(2) To persuate the advantages of using the proposed method, the authors should compare the results of using the proposed methods with other methods reported in literature.
As pointed out to our response to (1), the revised version of manuscript includes a comparison of the results obtained with several classifier methods popular in the ML literature, namely ANN, SVM, RF, LR, KNN and ADB, both with hyper-parameters and without hyper-parameters (cf. Tables 3 & 4 in the revised version). As a side contribution, this comparison serves to point out the need to refine results by means of hyper-parameters to stabilize results, again focusing on maximizing recall.
(3) Instead of descritive, it is suggested to use graphical means such as flow charts to descrive the proposed method and algorithm.
The revised version of the manuscript includes new content to graphically illustrate the algorithm’s operation (Figure 1 and Figure 2). An explanatory visual model was added, according to current XAI techniques. In particular, SHAP (SHapley Additive exPlanations) values were applied, which offer a high level of interpretability for a model (cf. new Figures 1 & 2). In other words, SHAP values are instrumental to show how much each predictor contributes, positively or negatively, to the target variable (i.e. cognitive state in our case: MCI, HC or AD).
- Importance of the characteristics: variables are ranked in descending order (from most to least important).
- Impact: The horizontal axis shows whether the effect of that value is associated with a higher or lower prediction.
- Original value: A color code indicates whether that variable is high (red) or low (blue) for that observation.
- Correlation: A high level of content labeled as timeDuration*REC-LP: long-term recognition duration time has a high and positive impact on the quality score. The high impact values are indicated in red, whereas positive impact is shown on the X axis. Similarly, we interpret that volatile acidity is negatively correlated with cognitive status.
(4) The quality of Figure 1 may need much improvement. Perhaps more statistic results can be presented in the manuscript to show variation, sensative, and accuracy of using the proposed method for the assessment.
The new version of the manuscript includes numerical evidence (cf. Tables 3 & 4) instead of the original Figure 1. In line with the reviewer’s suggestions, the new tables also include additional data not present in the original Figure 1.
2 Reviewer 2:
Manuscript report “Evaluation of the predictive ability and user acceptance of Panoramix 2.0, an AI‐based e‐health tool for the detection of cognitive impairment”, Works seems to be very much interesting and it has huge significance in detection of Dementia, I strongly recommend its publication. Few minor revisions are suggested
- Kindly increase the font size of Figure 1. (Idem 4 reviewer1)
Figure 1 has been replaced with new Tables 3 & 4. See response to our response to reviewer 1’s point (4) above.
- A scheme of the study need to be incorporated in the revised manuscript. (Idem 3 reviewer1)
See response to reviewer’s 1 point (3) above.
3 Reviewer 3:
The paper is devoted for Alzheimer-type dementia electronic testing methods. The topic is generally interesting, however the paper contains unexplained places and need major revisions.
In Abstract should be added more information about obtained results. The aim of the paper should be more clearly formulated.
The abstract of the revised manuscript has been extended according to the reviewer’s suggestion, and additional details have been included along the text and at the conclusion section to clarify the aim of this research.
More information about the digital neuropsyichological test should be added (part 2.2.4).
Section 2.2.4 was extended according to the reviewer’s suggestion.
Figure 1 and Tables 1-3 should be more commented.
Figure 1 was replaced by two new tables, which are more informative. All data has been further discussed along the text, as suggested.
Conclusions should be rewritten in more informative way.
Conclusions were rewritten as suggested.

Reviewer 2 Report
The work presenred in this manuscript is primary on using machine learning method for early detection of Alzheimer‐type dementia. The work can be included for possible publication in this journal due to the following reasons:
(1) Although the machine learning method used in the subject focus of interes is not new, it does fulfill the technology gap of traditional neuro-psychological tests, which are time consuming and inaccurate.
(2) The method and test procedure are clearly described and the concluding remarks are supported by the results, of which results indicate the work does meet archival journal merits of enhancement of exiting knowledge in the cognitive assessment method for the Alzheimer‐type dementia.
Where are areas for improvement:
(1) Although the present results are promising, it is suggested to investigate more on the sensitivity of information questionnaires and ML algorithms used in the digital assessment. Will the results be different with different algorithm such as decision tree, AI, ANN, etc?
(2) To persuate the advantages of using the proposed method, the authors should compare the results of using the proposed methods with other methods reported in literature.
(3) Instead of descritive, it is suggested to use graphical means such as flow charts to descrive the proposed method and algorithm.
(4) The quality of Figure 1 may need much improvement. Perhaps more statistic results can be presented in the manuscript to show variation, sensative, and accuracy of using the proposed method for the assessment.
Author Response

(The authors gave the same response as above.)

Reviewer 3 Report
Comments to author
Manuscript report “Evaluation of the predictive ability and user acceptance of Pan‐ oramix 2.0, an AI‐based e‐health tool for the detection of cogni‐tive impairment”, Works seems to be very much interesting and it has huge significance in detection of Dementia, I strongly recommend its publication. Few minor revisions are suggested
1. Kindly increase the font size of Figure 1.
2. A scheme of the study need to be incorporated in the revised manuscript.
Author Response

(The authors gave the same response as above.)

Reviewer 4 Report
The paper is devoted for Alzheimer-type dementia electronic testing methods. The topic is generally interesting, however the paper contain unexplained places and need major revisions.
In Abstract should be added more information about obtained results.
The aim of the paper should be more clearly formulated.
More information about the digital neuropsyichological test should be added (part 2.2.4).
Figure 1 and Tables 1-3 should be more commented.
Conclusions should be rewritten in more informative way.
Author Response

(The authors gave the same response as above.)

Round 2
Reviewer 1 Report
As none of my concern has been responded, I remain my last suggestion.
Author Response
Rebuttal (electronics-1933078):
Title:
Evaluation of the predictive ability and user acceptance of Panoramix 2.0, an AI-based e-health tool for the detection of cognitive impairment
Reviewer 4 (now Reviewer 1):
Authors are deeply sorry for not considering your comments and suggestions along the first revision cycle. This issue occurred because we incorrectly understood that we would receive a notification in case additional reviews were submitted after the three initial ones. We only knew about your review upon submission of the first revised version of the paper on the deadline indicated by the editors.
In any case, we are eager to respond to your concerns along the following paragraphs.
1 Authors didn’t mention why they particular interested in Panoramix 2.0?
The first version of Panoramix was also created and validated by the authors. This first version was composed of six different games addressing six different memory areas. Although psychometrically validated, the original version required a significantly greater amount of time and was more disruptive to the subjects analyzed. As a consequence of our experience administering Panoramix, we postulated that valid results may also be obtained with just three of the games by carefully selecting the most informative variables. This paper is the account and results of that research. This has been clarified in the revised version of the paper.
2 SD are missing in table 2 and 3.
SD values were unnoticedly deleted when formatting tables 2 and 3 for the journal’s Word template (tables 2 and 5 in the revised version of the paper). These values were recovered for the revised version.
3 Error bars and figure legends in figure 1 are missing.
Figure 1 was removed from the revised version of the paper and replaced by new tables 3 & 4 to include additional details about the ML-based analysis. Tables 3 & 4 include also SD values.
4 Abbreviations are used without definition.
Please accept our apologies for the missing definitions. The paper was thoroughly revised for abbreviations and acronyms not being defined the first time that they are used. The paper was also thoroughly revised for typos and grammar.

Reviewer 4 Report
Authors make proper corrections according to reviewer remarks and I suggest
publish the paper as it is.
Author Response
Thank you very much for your comments, which were extremely helpful to improve the quality of our paper